# Integration of Cellular and Humoral Immune Responses as an Immunomonitoring Tool for SARS-CoV-2 Vaccination in Healthy and Fragile Subjects

**DOI:** 10.3390/v15061276

**Published:** 2023-05-30

**Authors:** Giulia Brisotto, Marcella Montico, Matteo Turetta, Stefania Zanussi, Maria Rita Cozzi, Roberto Vettori, Romina Boschian Boschin, Lorenzo Vinante, Fabio Matrone, Alberto Revelant, Elisa Palazzari, Roberto Innocente, Giuseppe Fanetti, Lorenzo Gerratana, Mattia Garutti, Camilla Lisanti, Silvia Bolzonello, Milena Sabrina Nicoloso, Agostino Steffan, Elena Muraro

**Affiliations:** 1Immunopathology and Cancer Biomarkers Units, Department of Cancer Research and Advanced Diagnostics, Centro di Riferimento Oncologico di Aviano (CRO), IRCCS, 33081 Aviano, Italy; matteo.turetta@cro.it (M.T.); szanussi@cro.it (S.Z.); mrcozzi@cro.it (M.R.C.); rvettori@cro.it (R.V.); romina.boschian@cro.it (R.B.B.); asteffan@cro.it (A.S.); emuraro@cro.it (E.M.); 2Clinical Trial Office, Scientific Direction, Centro di Riferimento Oncologico di Aviano (CRO), IRCCS, 33081 Aviano, Italy; marcella.montico@cro.it; 3Division of Radiation Oncology, Centro di Riferimento Oncologico di Aviano (CRO), IRCCS, 33081 Aviano, Italy; lorenzo.vinante@cro.it (L.V.); fabio.matrone@cro.it (F.M.); alberto.revelant@cro.it (A.R.); elisa.palazzari@cro.it (E.P.); roberto.innocente@cro.it (R.I.); giuseppe.fanetti@cro.it (G.F.); 4Department of Medical Oncology, Centro di Riferimento Oncologico di Aviano (CRO), IRCCS, 33081 Aviano, Italy; lorenzo.gerratana@cro.it (L.G.); mattia.garutti@cro.it (M.G.); camilla.lisanti@cro.it (C.L.); silvia.bolzonello@cro.it (S.B.); 5Molecular Oncology Unit, Department of Cancer Research and Advanced Diagnostics, Centro di Riferimento Oncologico di Aviano (CRO), IRCCS, 33081 Aviano, Italy; mnicoloso@cro.it

**Keywords:** SARS-CoV-2, mRNA vaccine, vaccination boost, cellular immune response, humoral immune response, antibody, IFN-γ, cancer patients

## Abstract

Cellular and humoral immunity are both required for SARS-CoV-2 infection recovery and vaccine efficacy. The factors affecting mRNA vaccination-induced immune responses, in healthy and fragile subjects, are still under investigation. Thus, we monitored the vaccine-induced cellular and humoral immunity in healthy subjects and cancer patients after vaccination to define whether a different antibody titer reflected similar rates of cellular immune responses and if cancer has an impact on vaccination efficacy. We found that higher titers of antibodies were associated with a higher probability of positive cellular immunity and that this greater immune response was correlated with an increased number of vaccination side effects. Moreover, active T-cell immunity after vaccination was associated with reduced antibody decay. The vaccine-induced cellular immunity appeared more likely in healthy subjects rather than in cancer patients. Lastly, after boosting, we observed a cellular immune conversion in 20% of subjects, and a strong correlation between pre- and post-boosting IFN-γ levels, while antibody levels did not display a similar association. Finally, our data suggested that integrating humoral and cellular immune responses could allow the identification of SARS-CoV-2 vaccine responders and that T-cell responses seem more stable over time compared to antibodies, especially in cancer patients.

## 1. Introduction

T-cell-mediated immunity is the first arm of adaptive immunity activated in response to a viral infection and is required for the subsequent induction of a B-cell response and the development of a specific immunological memory lasting over time. Polyfunctional T-cell responses were reported in SARS-CoV-2 recovered individuals after infection [1], and a high correlation was observed between the neutralization antibody titer and the number of virus-specific T lymphocytes [2]. In particular, the presence of SARS-CoV-2-specific IFN-γ-producing CD4+ T helper (Th)1 cells and CD8+ cytotoxic T lymphocytes was associated with reduced disease severity, while CD4+ and CD8+ T-cell lymphopenia was revealed in severe COVID-19 cases [3]. The role of T-cell response becomes particularly relevant in patients affected by agammaglobulinemia or treated with B-cell-depleting agents, where T cells compensate for the lack of humoral immunity [3,4]. Moreover, while humoral response usually shows a progressive decline from one to eight months after SARS-CoV-2 infection, IFN-γ-secreting CD4+ T cells seem to last longer [5] and to develop even in the absence of seroconversion [6]. This observation is consistent with previous studies on other coronaviruses such as SARS-CoV. Indeed, in SARS-CoV infection, while the IgG antibody titer decreased dramatically one year after infection, the SARS-CoV-specific T-cell response persisted until 6 and even 11 years later, showing enhanced durability compared to antibody response [7,8]. Nevertheless, monitoring T-cell response in clinical practice appears more complex than antibody detection [3]. Thus, the development and validation of specific immunological assays are fundamental for defining the level and the duration of cellular immunity following SARS-CoV-2 infection, as well as for evaluating vaccination efficacy [9]. In this context, immunoinformatic analysis allowed the identification of the most immunogenic regions of the spike protein, able to stimulate the immune cell response in 80–100% of the global population [10,11]. IFN-gamma release assays (IGRAs) employing peptides derived from these immunogenic regions have been recently proposed to monitor cellular immunity after infection and/or vaccination [12,13]. Indeed, recent evidence reported that SARS-CoV-2 mRNA vaccines are able to elicit both high levels of spike-specific antibodies and strong Th1 functional CD4+ and CD8+ T cells [4,13]. As for the natural infection, evidence suggests that a robust T-cell response is required for the generation of high-titer neutralizing antibodies also after vaccination [3]. Vaccine-induced CD4+ and CD8+ T cells specific for SARS-CoV-2 are mainly central memory and effector memory T cells, similar to those developed after natural infection [4]. However, the duration of both the humoral and the cellular immune protection induced by vaccination in healthy subjects is still under evaluation. While an antibody decay was reported from 4 to 8 months after the second vaccination dose, independently of age and sex [14,15], memory B cells persist at least 6 months after vaccination [16], and cell-mediated immunity seems to last even longer [3]. However, current knowledge about factors mainly influencing the heterogeneity in immunological response is still limited [13]. The induction of effective cellular immunity against the virus after vaccination is particularly relevant in those subjects characterized by an immunodepression condition, which could compromise the vaccine efficacy and the development of an adequate antibody titer. The exclusion of immunodepressed subjects, such as transplant or cancer patients, from the main clinical studies on SARS-CoV-2 vaccines based on mRNA or adenoviral vector [17,18,19] did not allow the evaluation of the real efficacy of vaccination in these individuals. Subsequent studies reported low seroconversion rates especially in patients with hematological malignancies, patients with solid cancer, in subjects receiving stem cell or solid organ transplantation, and in patients affected by diseases requiring immunosuppressive therapies [20,21,22,23,24,25,26,27]. In this context, a parallel immunomonitoring of vaccine-induced cellular immunity could better define vaccination coverage in fragile patients. On these grounds, we designed the present observational prospective study to evaluate the T-cell-mediated immunity against the spike protein in healthcare workers (HCWs) and cancer patients who had received anti-SARS-CoV-2 vaccination at least 4 months prior to the study and to compare the cellular immune response with the antibody trend. We previously characterized a larger population of HCWs showing seroconversion in 99.9% of cases and a significant antibody decay 4 months after vaccination [15]. The main aim of the present study was to evaluate whether subjects characterized by a different antibody titer 4 months after the second vaccination dose showed similar rates of cellular immune response. Moreover, comparing the vaccine-induced cellular immunity in healthy subjects and cancer patients, we assessed whether the cancer-induced immunosuppression could impair vaccination efficacy.

## 2. Materials and Methods

### 2.1. Study Design and Sample Collection

This observational prospective study was conducted at the CRO Aviano, National Cancer Institute, IRCCS, to assess the T-cell immune response to the spike protein in subjects vaccinated with an mRNA SARS-CoV-2 vaccine and to compare the cellular and humoral immune response kinetics. For this purpose, subjects were divided based on their antibody titer in a first group including cases with a low level of IgG against SARS-CoV-2 spike-receptor binding domain (S-RBD) (i.e., the I tertile) and a second one comprising those with higher amounts of antibodies (i.e., the II and III tertiles). The sample size was calculated assuming that the frequency of positive cellular immune responses would have been higher in 66% (II and III tertiles) of subjects. We supposed that the expected difference between the two groups would have been of medium value (effect size h = 0.5). On these assumptions and considering a two-tailed z-test with α = 0.05 and β = 0.80, the expected calculated sample size was 141 subjects. Having considered a possible loss of samples due to technical issues, we aimed to enroll at least 150 subjects. The study included 2 cohorts: (i) healthcare workers (HCW, *n* = 156) of the CRO Aviano, National Cancer Institute, IRCCS and (ii) a convenience sample of cancer patients (*n* = 35) with solid tumor malignancies undergoing chemo- and/or radiotherapy treatment or a maximum of 2-year follow-up since the end of therapy. The inclusion criteria were as follows: age from 18 to 70 years and second dose of BNT162b2 or mRNA-1273 vaccine received at least 4 months before the enrollment. The exclusion criteria were as follows: (i) HCWs undergoing therapy or having less than 2-year follow-up after treatment for cancer disease, (ii) cancer patients with a hematological disease or receiving immunotherapy treatments, and (iii) individuals with HIV and/or HBV infection. At recruitment, all participants were interviewed to collect information about demographic data, time and type of SARS-CoV-2 vaccine received, type of vaccine side effects (fever > 38 °C, lymph node swelling, injection site pain/swelling, injection site itch, nausea/vomit/diarrhea, face paralysis, muscle/joint pain, fatigue, headache, allergic reaction/rash, mild symptoms), the presence of concomitant autoimmune or inflammatory disease and/or anti-inflammatory treatment at the time of vaccination or assessment of immune response, previous history of COVID-19 infection (monitored by swab test in HCWs and by self-declaration for cancer patients), and the level of SARS-CoV-2 S-RBD IgG (if available). In order to evaluate the kinetics and long-term persistence of the immune response, blood samples were collected at different time-points as specified below. Data were updated at each follow-up time-point. The study was approved by the Institutional Review Board (approval No. 10373/AG) and the National Ethics Committee of the Spallanzani IRCCS Institute (approval No. 464 of the trial register 2020/2021) and reported to the Regional Ethics Committee (Comitato Etico Unico Regionale del Friuli Venezia Giulia). All participants meeting eligibility criteria provided written informed consent for the conservation and use of biological samples, stored in the Institute Biobank.

HCWs’ blood samples for the humoral immune response evaluation were collected in 4.9 mL serum gel tubes at 3 time-points: 1 month after the second dose (IgG_T1), at least 4 months after the second dose (IgG_T2), and at least 4 months after the booster dose of mRNA vaccination (IgG_T3). 

HCWs’ blood samples for the cellular immune-mediated response were collected in 4.9 mL of lithium heparin at 2 time-points: at least 4 months after the second dose (QuantiFERON, QF_T2) and at least 4 months after the booster dose of mRNA vaccination (QF_T3).

Cancer patients’ blood samples were collected for both humoral and cellular immune response evaluation at least 4 months after the second dose of mRNA vaccination (IgG_T2 and QF_T2).

### 2.2. Assessment of SARS-CoV-2 Immune Responses

The cellular immune-mediated response was evaluated by using the QuantiFERON SARS-CoV-2 assay (Qiagen, Hilden, Germany), an interferon-gamma release assay (IGRA), which uses proprietary mixes of SARS-CoV-2-specific antigens (Ag) to stimulate lymphocytes in heparinized whole blood. This assay consists of two Ag tubes, SARS-CoV-2 Ag1 and SARS-CoV-2 Ag2, that contain, respectively, CD4+ T-cell epitopes from the S1 subunit (RBD) of the spike protein and both CD4+ and CD8+ T-cell epitopes derived from the S1 and S2 subunits of the spike protein. The QuantiFERON Nil and Mitogen collection tubes were used as negative and positive controls, respectively. Briefly, 4.9 mL of blood was drawn into lithium heparin blood tubes, and 1 mL of blood was dispensed for each QuantiFERON tube. Samples were gently mixed to resolubilize the Ag-content dried onto the inner walls and then incubated at 37 °C for 16–24 h. Plasma was harvested following centrifugation of QuantiFERON tubes for 15 min at 2000 RCF and used for the detection of IFN-γ levels by using the QuantiFERON ELISA assay (Qiagen, Hilden, Germany), according to the manufacturer’s recommendations. Following ELISA, results analysis was performed using the QuantiFERON Software. IFN-γ concentration was calculated by subtracting the value of the Nil tube. A positive response was considered for IFN-γ values ≥ 0.15 International Units (IU)/mL.

Plasma obtained from stimulated samples was harvested and stored at −80 °C for further measurement of interleukin-2 (IL-2) and tumor necrosis factor-α (TNF-α) using the automatic ELISA ELLA Protein Simple (Bio-Techne, Minneapolis, MN, USA) following the manufacturer’s instructions. IL-2 and TNF-α levels were calculated by subtracting the value of the Nil tube. 

The humoral immune response was evaluated as previously described [15]. Briefly, IgG levels against SARS-CoV-2 spike RBD were determined in serum using the MAGLUMI SARS-CoV-2 S-RBD IgG kit on a MAGLUMI 800 analyzer (Snibe Diagnostic, Shenzhen New Industries Biomedical Engineering Co., Ltd., Shenzhen, China) following the manufacturer’s instructions. The presence of reactivity was established for values ≥ 1.1 AU/mL, and the absence of reactivity was established for values < 0.9 AU/mL; values between 0.9 and 1.1 AU/mL were considered undetermined [28].

### 2.3. Statistical Analysis

Data were reported as frequencies and percentages for categorical variables and as median and interquartile range (IQR) for continuous variables. When required, continuous variables were dichotomized based on specific cut-offs (IFN-γ) or median values (IL-2 and TNF-α). Given the skewed distribution of values, the difference in IgG or IFN-γ values between groups was assessed with a Kruskal–Wallis or Mann–Whitney test, as appropriate. A Wilcoxon signed-rank test was used to evaluate the difference between paired observations. Correlation between continuous variables was assessed using Spearman’s correlation coefficient (rho). The test for trend across ordered groups developed by Cuzick was used to assess the trend of the number of vaccine side effects across IgG tertiles. Chi-squared test or Fisher’s exact test were used to evaluate the association between categorical variables, as appropriate, while McNemar’s test was used to assess change in time of dichotomic variables. Multivariate logistic regression was used to explore the independent role of variables in QF results. The IgG_T2 cut-off able to properly identify QF_T2 negative and positive subjects was determined by receiver operating characteristic (ROC) curve and Youden’s index analysis. In box-plot graphics, the horizontal line represents the median value, the box the interquartile range, and the whiskers the lower and higher values included in the following interval: 1st quartile − 1.5 × (3rd − 1st quartile) and 3rd quartile + 1.5 × (3rd − 1st quartile); values outside this interval are considered outliers (dots). A *p*-value < 0.05 was considered statistically significant. Data were analyzed by using Stata 14.2 software. 

## 3. Results

### 3.1. Study Population and Clinical Characteristics

Between November 2021 and April 2022, we enrolled a total of 156 healthcare workers (HCWs) and 35 cancer patients, whose clinical characteristics are summarized in Table 1. The two populations differed in age (HCWs median 44, IQR 34–54, versus patients median 56, IQR 49–64, *p* < 0.001) and the schedule of vaccination, which included two doses of the same mRNA-based vaccine, or one dose and a previous infection (*p* < 0.001 including the unknown cases). No differences were detected for sex, concomitant autoimmune diseases, anti-inflammatory treatments during vaccination or the evaluation of the immune response, or previous SARS-CoV-2 infection. Side effects were recorded after both vaccination doses in 140 HCWs and 23 patients, with no differences between groups. Interestingly, considering the global study population, an increase in the rate of side effects was observed after the second vaccination dose (McNemar’s test *p* = 0.019).

The immune response against SARS-CoV-2 was evaluated as antibody titer in 146 HCWs 30 days after the second vaccination dose (IgG_T1) and in 155 HCWs and 35 patients at least 120 days after the administration of two vaccine doses (IgG_T2). Moreover, 156 HCWs and 35 patients were tested for the presence of SARS-CoV-2-specific T cells through QF assay at least 120 days after the second vaccine dose (QF_T2). A subgroup of 50 HCWs was also characterized for the presence of a polyfunctional T-cell immune response. They were randomly selected to obtain two groups of 25 HCWs, respectively extracted from negative and positive responders at the QF_T2. The same analysis was performed also for all cancer patients. Finally, the subgroup of 50 selected HCWs was investigated for both humoral and cellular immune responses at least 120 days after the booster vaccine dose (IgG_T3 and QF_T3), and the relative side effects were reported.

### 3.2. Assessment of the SARS-CoV-2-Specific Humoral Immune Response

The specific humoral immune response against SARS-CoV-2 was evaluated by measuring the level of spike RBD-specific IgG using MAGLUMI technology. In HCWs, we recorded a median value of 568.2 AU/mL (IQR 361.1–924.1) and 98.7 AU/mL (IQR 70.4–179.4) for IgG_T1 and IgG_T2, respectively. We found a strong positive correlation between IgG_T1 and IgG_T2 values (Spearman’s rho 0.69, *p* < 0.001; Figure 1). Data referring to subjects with a previous history of SARS-CoV-2 infection show a similar distribution compared to infection-naïve HCWs (Figure 1).

The level of IgG_T2 for cancer patients was 47.9 AU/mL (IQR 13.2–299.6). However, we could not compare the amount of IgG_T2 between HCWs and patients (HCWs median 126 days, IQR 120–135; cancer patients median 177 days, IQR 156–203; *p* < 0.001), since the time elapsed from vaccination to antibody evaluation was different due to the time of enrollment. Overall, we observed only one HCW and one cancer patient with an IgG_T2 level under the cut-off value.

We then evaluated the possible correlation between the presence/absence of vaccination side effects after the second dose and antibody titer. As it could be expected, in HCWs, the IgG_T1 level was significantly higher in subjects reporting at least one side effect (no side effects, median 362.1 AU/mL, IQR 258.4–539.4; at least one side effect, median 728.6 AU/mL, IQR 420.6–976.1; *p* < 0.001; Table 2), and this trend was maintained also for IgG_T2 (no side effects, median 84.2 AU/mL, IQR 53.3–99.2; at least one side effect, median 118.6 AU/mL, IQR 74.5–250.5; *p* < 0.001; Table 2). Interestingly, by stratifying HCWs based on IgG_T2 levels, we noticed that the reported side effects were higher in the II and III antibody tertiles (*p* for trend = 0.017; Figure 2). Finally, HCWs with a previous history of SARS-CoV-2 infection showed a higher amount of IgG_T2 compared to infection-naïve subjects (no infection, median 94.3 AU/mL, IQR 70.0–161.8; previous infection, median 248.1 AU/mL, IQR 130.1–868.0; *p* < 0.001; Table 2), in accordance with previous evidence [13]. Conversely, no difference was detected when evaluating the presence/absence of autoimmune diseases and anti-inflammatory therapies taken both throughout vaccination and during immune response evaluation (Appendix A).

Unfortunately, the partial lack of information regarding patients’ reported side effects after vaccination or previous history of SARS-CoV-2 infection did not allow a reliable statistical analysis between these parameters and IgG_T2 levels. Autoimmune diseases and anti-inflammatory therapies did not influence the humoral response in patients (Appendix A).

No differences were observed in HCWs or cancer patients based on sex (Appendix A).

### 3.3. Evaluation of the SARS-CoV-2-Specific Cellular Immune Response

The specific cellular immune response against SARS-CoV-2 was evaluated by detecting the presence of spike-specific T lymphocytes through the QF SARS-CoV-2 IGRA after at least 4 months from the end of the vaccination schedule (QF_T2). The results of this assay were considered both as qualitative (negative/positive with respect to the 0.15 IU/mL IFN-γ threshold) and quantitative data (levels of IFN-γ IU/mL after Ag1 or Ag2 stimulation). 

We detected a positive QF_T2 response in 64.1% of the HCWs, contrary to cancer patients, which showed a positive response only in 40.0% of cases (*p* = 0.013). This difference was still significant after adjusting for age in multivariate logistic regression, while age was not associated with QF_T2 response (adjusted *p* = 0.602). Both the cases with an IgG_T2 titer under the cut-off of 1.1 AU/mL (one HCW and one cancer patient) showed a negative QF_T2 response. Intriguingly, in the stratified analysis, patients with a positive response were younger than those characterized by a negative result (*p* = 0.045), while no difference based on age (*p* = 0.853) or sex (*p* = 0.661) was observed in the HCW cohort. As for IgG_T2 determination, we noticed a significant difference in the time elapsed from the second vaccination dose and the QF_T2 evaluation between HCWs and cancer patients (HCWs median 126 days, IQR 120–135; cancer patients median 177 days, IQR 156–203, *p* < 0.001). However, considering qualitative QF_T2 results, this time was not different between cases showing a negative or a positive response to QF, either in HCWs (*p* = 0.128) or in cancer patients (*p* = 0.476), suggesting that QF results could be more stable over time compared to IgG levels. Thus, we evaluated QF also as quantitative results and observed that IFN-γ levels were higher both after Ag1 (*p* = 0.020) and Ag2 (*p* = 0.018) stimulation in HCWs compared to cancer patients (Figure 3A,B). To characterize the polyfunctionality of cellular immune response, we also quantified IL-2 and TNF-α as T-cell-specific cytokines in a cohort of 50 HCWs and in all cancer patients. Interestingly, we noticed that TNF-α levels were higher in the HCW cohort compared to cancer patients both after Ag1 (*p* < 0.001) and Ag2 (*p* < 0.001) stimulation (Figure 3C,D). Further, by investigating the HCW cohort, we found an evident correlation between IFN-γ results measured through QF and IL-2 concentrations considering either Ag1 (Spearman’s rho = 0.822, *p* < 0.001) or Ag2 stimulation (Spearman’s rho = 0.859, *p* < 0.001; Appendix A). Similar results were observed in cancer patients (Ag1, Spearman’s rho = 0.847, *p* < 0.001; Ag2, Spearman’s rho = 0.862, *p* < 0.001; Appendix A).

In HCWs, the presence of vaccination side effects (after first and/or second dose) was associated with a positive QF_T2 response (*p* = 0.002), independently of age. Similarly, a higher frequency of side effects was detected in the case of increased IL-2 levels both after Ag1 (*p* = 0.027) and Ag2 (*p* = 0.027) stimulation, compared to low IL-2 levels, while no association was detected for TNF-α values (Ag1, *p* = 0.951; Ag2, *p* = 0.362). Furthermore, a higher frequency of QF_T2 positive results was observed for HCWs with a previous history of SARS-CoV-2 infection (*p* = 0.019). Like the humoral response, cellular immunity was not affected by the presence/absence of autoimmune diseases (*p* = 0.606) and anti-inflammatory therapies taken throughout vaccination (*p* = 1.000) and during immune response evaluation (*p* = 0.757).

By stratifying HCWs for their qualitative QF_T2 data, we observed significantly higher IgG_T1 and IgG_T2 levels in case of a positive response after Ag1 or Ag2 stimulation, and as global QF_T2 results (i.e., being positive for at least one Ag; Table 3). Of note, the differences were still significant if considering only SARS-CoV-2 infection-naïve subjects. Similar results, even if not significant, were observed in the cancer patient cohort for the IgG_T2 levels (Table 3).

By grouping HCWs in tertiles based on IgG_T1 levels, we observed that subjects in the II and III tertiles showed a higher frequency of QF_T2 global positive response (76.3%) compared to those characterized by a lower level of IgG_T1 (23.7%; *p* = 0.024). Similar results were also detected considering the response to single antigens, even if it was significant only for Ag1 (Ag1 80.5% versus 19.5%, *p* = 0.003; Ag2 75.6% versus 24.4%, *p* = 0.057).

By multivariate logistic regression analysis, we found that belonging to the II and III tertiles of both IgG_T1 (for HCWs) and IgG_T2 (for HCWs and cancer patients) increased the probability of a positive QF_T2 response (Table 4). Consistently, HCWs of the II and III IgG_T2 tertiles showed a higher level of IFN-γ, both after Ag1 and Ag2 stimulation, compared to subjects of the I IgG_T2 tertile (Appendix A). Interestingly, this difference was also maintained in HCWs not experiencing a previous SARS-CoV-2 infection (Appendix A). 

Assessing the fold change of IgG levels, we detected a more evident decay in IgG_T2 titer with respect to IgG_T1 amount in QF_T2 negative HCWs compared to subjects with a detectable cellular immunity (*p* = 0.031; Figure 4).

In order to define an antibody titer (IgG_T2) able to properly identify QF_T2 negative and positive subjects, we performed a ROC curve analysis with a training population of 78 cases (AUC = 0.705; Appendix A). Through Youden’s index analysis, we found a cut-off of 99.2 AU/mL IgG_T2 with a sensitivity and a specificity of 62.5% and 76.7%, respectively. In the validation set (*n* = 112), the ROC curve showed a sensitivity of 54.5% and a specificity of 65.2%, correctly identifying 58.9% of cases.

### 3.4. Evaluation of the SARS-CoV-2-Specific Humoral and Cellular Immune Response after Vaccination Boost in HCWs

Fifty HCWs were included for further analysis of IgG_T3 and QF_T3 response after the booster vaccine dose. Globally, we observed a median IgG_T3 value of 779.0 AU/mL (IQR 345.7–1704.0) and a positive QF_T3 response in 60.0% of cases. Interestingly, all cases showing a positive response to QF_T2 maintained a positive response after the booster vaccine dose, while we found a shift in 20% of QF_T2 negative cases, which became QF_T3 positive. Intriguingly, we noticed a strong correlation between QF_T3 and QF_T2 quantitative levels both after Ag1 and Ag2 stimulation (Figure 5A,B; Ag1 Spearman’s rho 0.768, *p* < 0.001; Ag2 Spearman’s rho 0.805, *p* < 0.001).

Both humoral immunity and cellular immunity were not affected by the presence/absence of autoimmune diseases (Appendix A for IgG_T3; QF_T3, *p* = 0.381) and anti-inflammatory therapies taken throughout vaccination (Appendix A for IgG_T3; QF_T3, *p* = 1.000).

## 4. Discussion

In the present observational prospective study, we monitored both the humoral and the cellular immune responses against the SARS-CoV-2 spike protein in HCWs and cancer patients at least 4 months after the end of the vaccination schedule (i.e., after two vaccine doses). Subject enrollment started in November 2021 when COVID-19 incidence in Italy was increasing weekly, with 112/100,000 cases [29], the most prevalent SARS-CoV-2 variant was Delta [30], and 84.5% of the population over 12 years old was fully vaccinated [31,32]. Our main aim was to evaluate whether the antibody level was associated with the T-cell response. In HCWs, we found that subjects belonging to the II and III highest tertiles of antibodies measured both 1 (IgG_T1) and 4 months after the end of the vaccination schedule (IgG_T2) had an increased probability of showing a positive cellular immune response (QF_T2). Similarly, in a subgroup of HCWs, also tested 4 months after the vaccination boost, higher levels of antibodies were observed in cases showing a positive T-cell response compared to subjects characterized by the absence of SARS-CoV-2-specific cellular immune response. Moreover, the negative antibody responses found in one case for each cohort (HCWs and cancer patients) were both characterized by the absence of cellular immunity. This observation is clearly preliminary, but it is consistent with previous analyses reporting a minority of cases negative for both types of immune response [24]. Interestingly, our analyses underlined that the presence of cellular immunity after vaccination was associated with a less pronounced antibody decay compared to a negative QF response. These data suggest a strong correlation between humoral and cellular immune response after COVID-19 vaccination, as recently reported in other healthy Italian and European subjects [14,24], and indicate that the presence of a T-cell response could be associated with a long-term B-cell response [33]. 

The onset of post-vaccination side effects with SARS-CoV-2 mRNA vaccines was commonly reported, particularly after the second dose [17,18,34,35]. Consistently, we observed the same trend in our study population, without detecting differences between the cohorts of HCWs and cancer patients. Regarding the effect of the vaccination boost, several works described a frequency of adverse reactions similar to those identified after the second, affecting about 70–80% of vaccines [36,37]. In this study, the percentage of individuals reporting at least one side effect after the third dose was slightly lower both compared to the literature evidence and the percentage detected after the second dose. However, we can not exclude that these discrepancies could be due to the lower number of individuals investigated after boosting. Of note, none of the individuals included in this study reported life-threatening side effects but rather reported side effects ranging from none to mild, thus corroborating the safety of the mRNA-based vaccination.

The relation between the manifestation of adverse reactions and the extent of the immune response [38,39,40,41] is also a matter of interest. At present, this is still debated due to the presence of conflicting results. For instance, few studies reported no association between post-vaccination reactions and IgG levels after vaccination [39,42,43], while others suggested a correlation between the extent of vaccination side effects and a higher antibody level [44,45,46,47,48]. Similarly, the underlying association between side effects and cellular response is still mainly unexplored and inconsistent [49,50,51,52]. In our cohort, we observed that the occurrence of adverse reaction was associated with an enhanced humoral response both at 1 and 4 months after the second vaccination dose in HCWs, and the same difference was maintained also for the subgroup of HCWs included for the immunity assessment after the vaccination boost. Intriguingly, the frequency of HCWs reporting more than one side effect was higher in the group of those with higher antibody levels (II-III tertiles of IgG_T2). Consistently, in the presence of post-vaccination adverse reactions, we detected a high level of cellular response, both in terms of IFN-γ results and IL-2 levels. Such an analysis, unfortunately, could not be performed for the cancer patient cohort because of a lack of information about the reactogenicity of the vaccination dose. Overall, our analysis suggests that the presence of vaccination side effects might reflect a stronger immune response, but further investigations are needed to confirm such results. Indeed, some limitations need to be taken into account: first, the side effects were self-reported and collected as a survey, and thus no objective evaluation could be performed; second, the survey was given to participants at least 4 months after the second dose, introducing the potential for recall bias. Finally, we did not record the severity and the extent of the reactions. 

When evaluating the cohort of cancer patients, we could not observe the same correlation between antibodies (IgG_T2) and T-cell responses (QF_T2) highlighted in HCWs, but we noticed a similar trend. The lack of significance could reflect the immune-suppressed status characterizing these fragile patients. Even if we included only a limited number of patients, the same assumption was reported also in larger cohorts of cancer patients where no significant or just a weak correlation was observed between anti-RBD antibodies and T-cell responses in fragile patients [24,53]. Interestingly, Bordry et al. reported that T-cell activation was more likely to occur in cancer patients experiencing seroconversion compared to those characterized by the absence of virus-specific antibodies [54]. They mainly observed a lack of seroconversion in patients affected by a hematological malignancy, which actually were excluded by our analysis, while in the presence of solid cancer, antibody response was evident in 97.2% of patients, which is consistent with our data [54]. The same observation was also reported by the CAPTURE study, enrolling a huge number of cancer patients and showing a lower seroconversion in patients affected by hematological malignancies, while reporting that 80% of patients showed a vaccine-induced T-cell response independently of cancer type [55]. Similarly, Bergamaschi et al. described lower humoral and reduced innate cytokine responses to vaccination in a cohort of patients affected by hematological malignancies and receiving bone marrow transplantation or CAR-T cells compared to a cohort of HCWs [56].

Literature evidence reported an impaired immune proficiency, especially in the humoral arm, in the presence of autoimmune diseases under treatment [57,58,59]. Our analysis included a low number of individuals characterized by these clinical conditions; thus, we might not reach final conclusions in this regard. Interestingly, we observed a significantly lower percentage of cancer patients respect to healthy subjects showing a positive T-cell response after vaccination. We also noticed that the amounts of IFN-γ and TNF-α released after stimulation with both the peptide mixes employed, the CD4+ T cell-stimulating Ag1 and the CD4+ and CD8+ T cell-stimulating Ag2, were higher in HCWs compared to cancer patients, thus suggesting a general impairment of the Th1 cell compartment in patients, as reported by others [24,60]. Unfortunately, we could not perform the same comparison for antibody levels, due to the different elapsed time frame between vaccination and IgG_T2 evaluation in HCWs and patients. Indeed, the demonstrated physiological decay of the SARS-CoV-2-specific humoral response after vaccination could impair this analysis [13,15]. 

It should be noted that HCWs and patients differed in their age and vaccination schedule, which, however, was unknown in one-third of patients. We hypothesized that all cancer patients likely underwent the same vaccination plans reported for HCWs due to the common Italian policy for vaccination administration. Within the cohort of cancer patients, age seemed to affect the development of a SARS-CoV-2-specific cellular immune response since younger subjects were more likely to show a positive response to QF compared to older ones. Divergent results have been previously reported in this regard, with some studies demonstrating an impairment of T-cell responses after SARS-CoV-2 mRNA vaccination in elderly people [3] and others showing no correlation between age and T-cell responses, contrarily to antibody responses, which decrease with age [4]. We reported an inverse correlation between age and antibody titer in a larger cohort of HCWs [15], while here the HCWs population did not show any difference for T-cell responses based on age. These results could imply a reduced T-cell responsiveness to COVID-19 vaccination in the presence of a general immunosenescence, usually more frequently observed in elderly subjects and in the presence of malignancies [3]. 

Both the methodologies used to evaluate the humoral and the cellular immune response specific for SARS-CoV-2 are not able to distinguish between vaccine- or infection-induced immunity [13]. However, we noticed that a previous history of SARS-CoV-2 infection was associated with higher levels of antibodies and also with a higher probability of a positive response to the QF assay in the HCW cohort investigated, as reported in other studies [13], thus suggesting that the combination of vaccination and natural infection could induce a better immunization than two vaccine doses or natural infection alone [14,33,61]. This observation could be useful in the future COVID-19 vaccination schedule definition, especially for countries or periods with scarce vaccine supplies [13]. 

By assessing factors associated with the persistence of a durable immune response after vaccination, we detected a strong correlation between IFN-γ results and the levels of IL-2, a cytokine essential for the homeostatic maintenance of a functional specific T-cell response [60]. Intriguingly, when comparing the SARS-CoV-2-specific immunity developed after the initial vaccination schedule and following the booster dose, we observed a strong correlation in the levels of IFN-γ released after stimulation with both Ag1 and Ag2 peptide mixes. Conversely, the respective antibody levels (IgG_T2 and IgG_T3) did not seem to be correlated. We only noticed that cases reporting the absence of cellular immunity at both time-points also showed a reduced amount of antibodies over time. This could imply a higher stability of the cellular immune response compared to the humoral one, which appeared more affected by fluctuations. Interestingly, also after SARS-CoV-2 infection, the virus-specific T-cell response persists longer than the antibodies that usually decay [62]. Moreover, in cancer patients, antibody titers seemed to offer only an incomplete picture of the vaccine-elicited SARS-CoV-2-specific immunity, while cellular immunity was detected also in the absence of significant quantities of specific antibodies [63,64].

## 5. Conclusions

Based on these data, a combined analysis of the humoral and cellular immune response is probably necessary for the real identification of SARS-CoV-2 vaccine responders [3,65], with particular attention to the evaluation of virus-specific T-cell responses especially in fragile cases, such as cancer patients, since they could be more stable over time compared to the levels of antibodies.

## Figures and Tables

**Figure 1 viruses-15-01276-f001:**
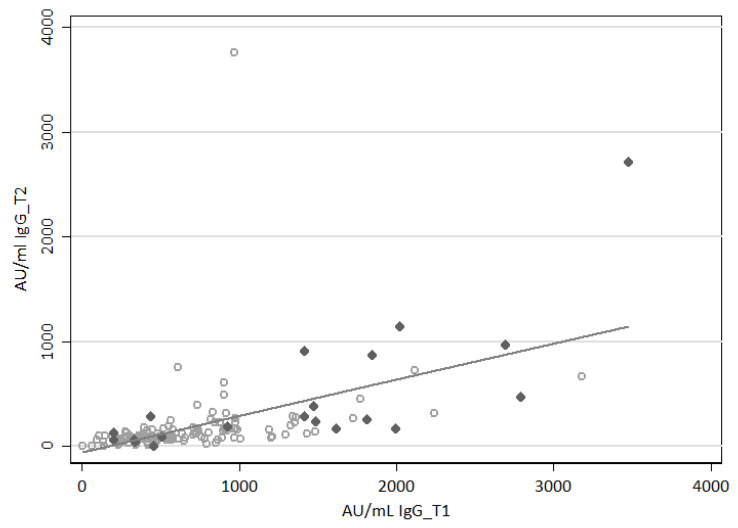
Dynamics of anti-SARS-CoV-2 IgG levels in paired HCW samples after the second vaccination dose. Correlation between anti-SARS-CoV-2 IgG levels evaluated at 30 (IgG_T1) and 120 days (IgG_T2) after the second dose of vaccination. Each symbol represents an individual subject whose antibody titer was available for both time-points (*n* = 146); empty circles refer to infection-naïve HCWs (*n* = 126), grey diamonds to individuals with a previous history of SARS-CoV-2 infection (*n* = 20); the solid line represents the linear trend; Spearman’s rho 0.69, *p* < 0.001. AU, arbitrary units.

**Figure 2 viruses-15-01276-f002:**
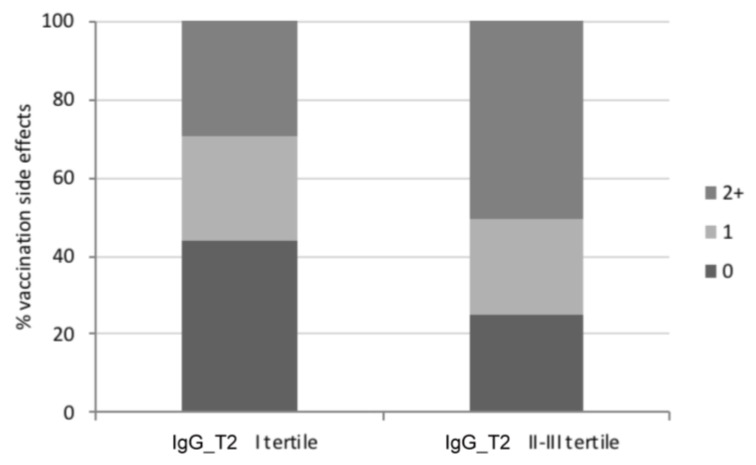
Frequency of the magnitude of HCWs’ vaccination side effects based on anti-SARS-CoV-2 IgG levels detected at least 120 days after the second vaccination dose (IgG_T2). By considering IgG_T2 distribution in tertiles, HCWs were divided into those with a low level of IgG (i.e., within the I tertile; IgG_T2 I tertile) and those with a higher amount of antibodies (i.e., within the II and III tertiles; IgG_T2 II-III tertile) and then grouped by the number of self-reported vaccination side effects (no side effects (0), dark grey; at least one side effect (1), light grey; ≥2 side effects (2+), grey); *p* for trend = 0.017 (McNemar’s test).

**Figure 3 viruses-15-01276-f003:**
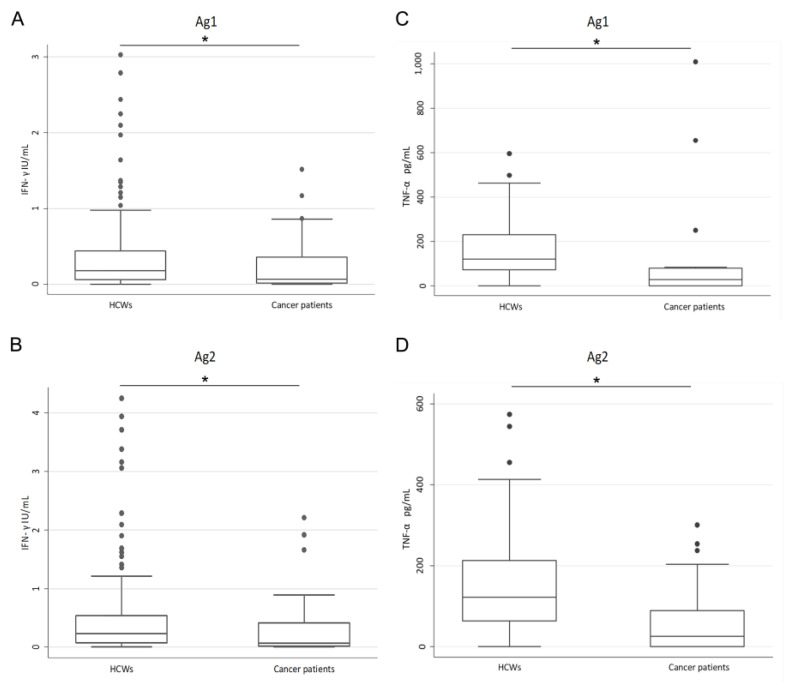
SARS-CoV-2-specific cellular immune response between HCWs and cancer patients. Comparison of the IFN-γ (**A**,**B**) and TNF-α (**C**,**D**) levels after Ag1 (**A**,**C**) and Ag2 (**B**,**D**) stimulation in HCWs and cancer patients, as detected at least 120 days after receiving 2 doses of vaccination. HCWs, healthcare workers; IU, international units. * *p* < 0.05.

**Figure 4 viruses-15-01276-f004:**
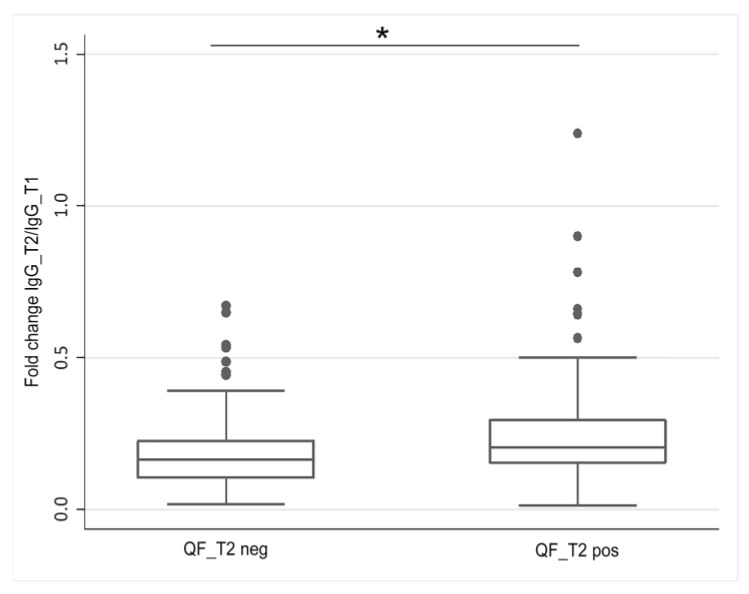
Dynamic of the anti-SARS-CoV-2 antibody levels with respect to SARS-CoV-2-specific cellular immunity status in HCWs. Comparison of the SARS-CoV-2 IgG decay, reported as the fold change of the IgG level at 120 (IgG_T2) and 30 days (IgG_T1) after the second vaccination dose, between HCWs resulting negative (QF_T2 neg) or positive (QF_T2 pos) for the presence of SARS-CoV-2-specific cellular immunity at least 120 days after the second vaccination dose.* *p* < 0.05.

**Figure 5 viruses-15-01276-f005:**
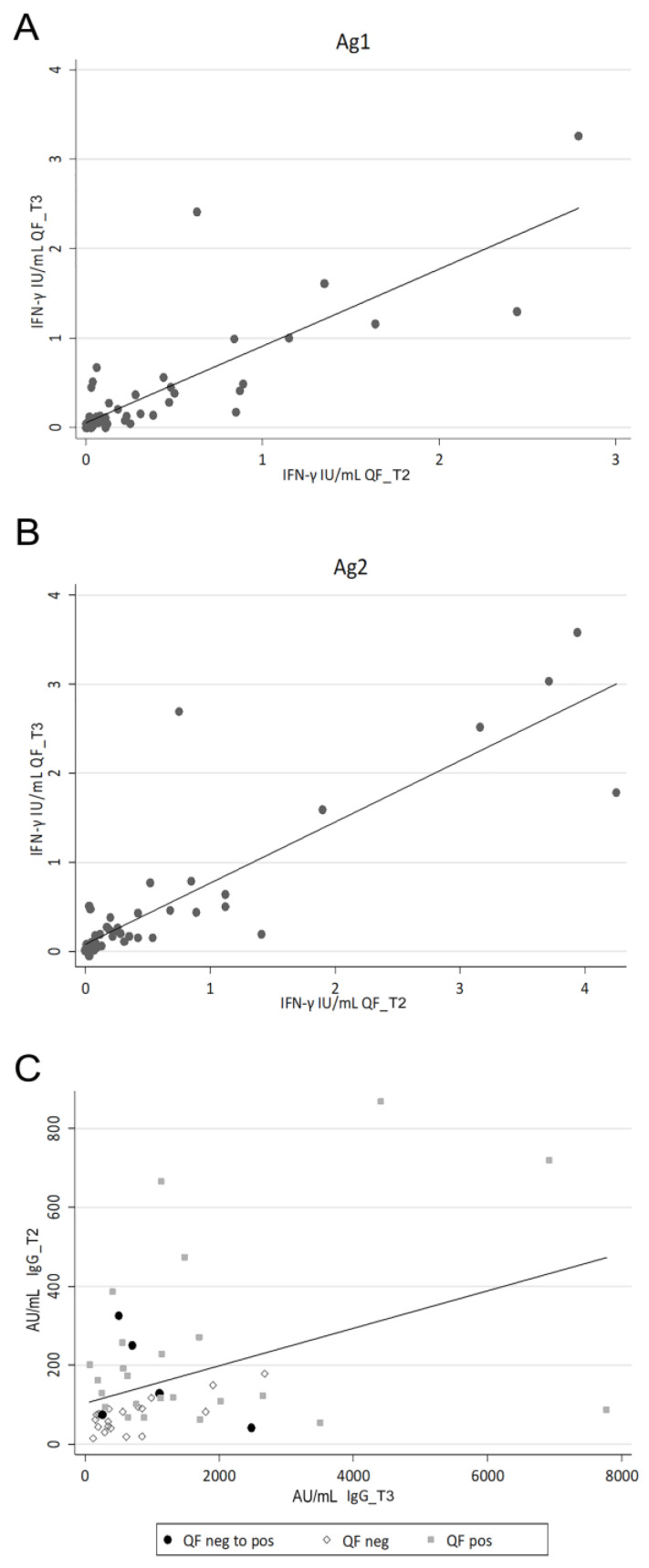
SARS-CoV-2-specific immunity before and after vaccination boost in HCW cohort. Correlation between IFN-γ levels evaluated at 120 days after the second (QF_T2) and the booster (QF_T3) vaccination dose after Ag1 (**A**) and Ag2 (**B**) stimulation. Correlation between specific SARS-CoV-2 IgG levels detected at 120 days after the second (IgG_T2) and the booster (IgG_T3) vaccination dose in HCWs stratified for their QF_T2 and QF_T3 global result status (**C**) (HCWs negative for QF_T2 and positive for QF_T3 were indicated as full black circles, HCWs negative for both identified as empty diamonds, HCWs positive for both QF_T2 and QF_T3 shown as full grey squares; the solid line represents the linear trend; Spearman’s rho 0.258, *p* = 0.070).

**Table 1 viruses-15-01276-t001:** Clinical characteristics of the enrolled population.

	HCWs	Cancer Patients	*p*-Value *
**Age**, median (IQR)	44 (34–54)	56 (49–64)	*p* < 0.001
**Sex**, *n* (%)			
Female	101 (64.7)	24 (68.6)	*p* = 0.667
Male	55 (35.3)	11 (31.4)	
**Vaccination schedule**, *n* (%)			
Only BNT162b2	140 (89.7)	20 (57.1)	*p* < 0.001
Only mRNA-1273	6 (3.9)	1 (2.9)	
Infection and mRNA vaccination	10 (6.4)	3 (8.6)	
Unknown	0 (0.0)	11 (31.4)	
**Autoimmune diseases**, *n* (%)			
No	139 (89.1)	34 (97.1)	*p* = 0.204
Yes	17 (10.9)	1 (2.9)	
**Anti-inflammatory therapies during vaccination**, *n* (%)			
No	148 (94.9)	31 (88.6)	*p* = 0.238
Yes	8 (5.1)	4 (11.4)	
**Anti-inflammatory therapies during immune response evaluation**, *n* (%)			
No	144 (92.3)	32 (91.4)	*p* = 0.741
Yes	12 (7.7)	3 (8.6)	
**Previous SARS-CoV-2 infection**, *n* (%)			
No	134 (85.9)	31 (88.6)	*p* = 0.791
Yes	22 (14.1)	4 (11.4)	
**Vaccination side effects**, *n* (%)			
None	31 (22.1)	11 (47.8)	*p* = 0.072
Only after 1st dose	11 (7.9)	2 (8.7)	
Only after 2nd dose	27 (19.3)	2 (8.7)	
After both doses	71 (50.7)	8 (34.8)	

* Mann–Whitney test for age; Fisher’s exact test for all the other variables. Abbreviations: HCWs, healthcare workers; IQR, interquartile range.

**Table 2 viruses-15-01276-t002:** Humoral response in HCWs and association with clinical parameters.

	IgG Levels (AU/mL)	*p*-Value *
**IgG_T1**, median (IQR)	568.2 (361.1–924.1)	-
**IgG_T2**, median (IQR)	98.7 (70.4–179.4)	-
**IgG_T3**, median (IQR)	779.0 (345.7–1704.0)	-
**Vaccination side effects**, median IgG_T1 (IQR)		
No	362.1 (258.4–539.4)	*p* < 0.001
Yes	728.6 (420.6–976.1)	
**Vaccination side effects**, median IgG_T2 (IQR)		
No	84.2 (53.3–99.2)	*p* < 0.001
Yes	118.6 (74.5–250.5)	
**Vaccination side effects**, median IgG_T3 (IQR)		
No	575.8 (290.9–852.7)	*p* = 0.032
Yes	1048.3 (594.1–1750.5)	
**Previous SARS-CoV-2 infection**, median IgG_T2 (IQR)		
No	94.3 (70.0–161.8)	*p* < 0.001
Yes	248.1 (130.1–868.0)	

* Mann–Whitney test. Abbreviations: AU, arbitrary units; IQR, interquartile range; d, days after second vaccination dose; dB, days after the boost vaccination dose.

**Table 3 viruses-15-01276-t003:** Cellular and humoral response in HCWs and cancer patients.

	QF Qualitative Results		
	Global	Ag1	Ag2
**IgG_T1**, median (IQR) *			
** *HCWs* **			
QF_T2 Neg	419.8 (304.5–736.0)	426.3 (298.8–735.3)	452.0 (315.8–751.6)
QF_T2 Pos	705.9 (415.4–1193.1)	778.6 (453.1–1355.0)	642.0 (412.6–1193.1)
*p-value*	*p < 0.001*	*p < 0.001*	*p < 0.001*
** *Infection-naïve HCWs* **			
QF_T2 Neg	432.7 (298.8–736.0)	436.0 (290.7–732.5)	458.4 (304.5–741.0)
QF_T2 Pos	609.0 (389.8–970.4)	677.8 (415.4–976.1)	579.1 (389.4–970.1)
*p-value*	*p = 0.004*	*p < 0.001*	*p = 0.015*
**IgG_T2**, median (IQR)*			
** *HCWs* **			
QF_T2 Neg	81.3 (44.5–139.2)	84.7 (47.1–130.1)	81.9 (45.0–139.2)
QF_T2 Pos	125.6 (85.2–247.0)	146.0 (85.7–274.0)	129.8 (85.2–248.1)
*p-value*	*p < 0.001*	*p < 0.001*	*p < 0.001*
** *Infection-naïve HCWs* **			
QF_T2 Neg	81.6 (44.0–133.7)	84.5 (47.1–128.1)	82.4 (45.5–133.7)
QF_T2 Pos	114.9 (80.5–173.2)	118.4 (80.5–228.1)	116.9 (78.3–173.2)
*p-value*	*p = 0.004*	*p < 0.001*	*p = 0.002*
** *Cancer patients* **			
QF_T2 Neg	30.4 (12.6–123.3)	36.3 (12.9–102.7)	30.4 (12.6–123.3)
QF_T2 Pos	79.0 (41.3–715.2)	126.4 (36.2–935.1)	79.1 (41.3–715.2)
*p-value*	*p = 0.069*	*p = 0.082*	*p = 0.069*
**IgG_T3**, median (IQR) *			
** *HCWs* **			
QF_T3 Neg	475.3 (250.1–850.3)	569.6 (290.9–852.7)	475.2 (256.1–847.9)
QF_T3 Pos	1120.0 (555.1–2018.1)	1136.1 (633.1–1711.5)	1132.6 (562.3–2249.9)
*p-value ***	*p = 0.021*	*p = 0.036*	*p = 0.008*
** *Infection-naïve HCWs* **			
QF_T3 Neg	475.3 (250.1–850.3)	569.6 (290.9–852.7)	475.3 (256.1–847.9)
QF_T3 Pos	1120 (569.6–2018.1)	1136.1 (638.9–1711.5)	1132.6 (601.4–2249.9)
*p-value ***	*p = 0.027*	*p = 0.042*	*p = 0.010*

* IgG levels are reported in AU/mL; ** Mann–Whitney test. Abbreviations: QF, QuantiFERON; Ag, antigen; HCWs, healthcare workers; d, days after the second vaccination dose; dB, days after the boost vaccination dose; neg, negative; pos, positive; IQR, interquartile range.

**Table 4 viruses-15-01276-t004:** Multivariate logistic regression analysis of QF_T2 qualitative results and IgG tertile levels adjusted for age and stratified by HCWs and cancer patients.

	Adjusted Odds Ratio	95% CI	*p*-Value
**IgG_T1 (HCWs)**			
I tertile	ref		
II-III tertiles	2.29	1.10–4.73	*p = 0.026*
Age	0.99	0.97–1.03	*p = 0.894*
**IgG_T2 (HCWs)**			
I tertile	ref		
II-III tertiles	3.81	1.70–8.53	*p = 0.001*
Age	1.01	0.98–1.04	*p = 0.429*
**IgG_T2 (Cancer patients)**			
I tertile	ref		
II-III tertiles	2.41	0.56–10.44	*p = 0.239*
Age	0.92	0.84–1.01	*p = 0.085*

Abbreviations: HCWs, healthcare workers; d, days after the second vaccination dose; dB, days after the boost vaccination dose.

## Data Availability

The data presented in this study are available on request from the corresponding author.

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
