# Peer review of "Integration of Cellular and Humoral Immune Responses as an Immunomonitoring Tool for SARS-CoV-2 Vaccination in Healthy and Fragile Subjects"

_viruses, 2023, doi:10.3390/v15061276_

Round 1

Reviewer 1 Report (Previous Reviewer 4)

Completed and well presented work.

Minor editing of English required for a more fluent reading.

Author Response

Thank you for your suggestion. We checked and corrected english language. 

Reviewer 2 Report (Previous Reviewer 3)

I confirm my previous judgement, it deals of a robust study (the second version improves the first one) providing a practically relevant contribution for the prevention of COVID-19   

Author Response

Thank you. 

Reviewer 3 Report (Previous Reviewer 1)

The resubmitted article has been substantially improved. However, still, the answers to a few questions are not satisfactory. I encourage/recommend adding these suggestions to make it more interesting. 

Though the author agrees on a higher no.of available data and referring Zou et al NEJM 2022 reported only after the bivalent vaccine whereas their data from the first-generation vaccine is not totally correct. Because many reports after the 2 doses of mRNA vaccines are published including cellular immune response as a biomarker in patients (https://www.ncbi.nlm.nih.gov/pmc/articles/PMC8669666/). It is totally understandable that their cohorts/analysis is different.

1. Including Covid-19 patients under health cohorts is not scientifically correct, according to me, hence consider revising the cohort name as "naive vaccine recipients". Also, show the 10 infected individuals in Fig 1 either by different colors or shapes to completely understand the data that might be interesting to the readers. The author provided suppl table 3 showing infected and uninfected but still, the main table has these subjects, modifying Fig 1 would be straightforward.

2. Author mentioned they couldn't compare IgG_T2 between HCW and cancer patients due to the time diff in enrollment. But within cancer cohorts, doing a correlation plot b/w IgG_T1 and T2 is still possible. Please add them. 

Author Response

  1. We greatly thank you for this interesting observation. As suggested, we modified  Figure 1 highlighting subjects with a previous history of SARS-CoV-2 infection with a different shape compared to infection-naïve HCWs. Similarly, we added the results referring to infection-naïve subjects to the main table 3.
  2. We agree with your kind suggestion, but, unfortunately, for the cancer patient cohort we do not have data about IgG_T1 to compare with IgG_T2.

This manuscript is a resubmission of an earlier submission. The following is a list of the peer review reports and author responses from that submission.

Round 1

Reviewer 1 Report

This article describes assessing both humoral and cellular immune responses to monitor SARS-CoV2 vaccine responses. It is not well written and also the study design should have been different. Their conclusion, high Ab titers in HCW, doesn't purely come from healthy cohorts but also from donors with an infection history.

Findings from this current work have already been published/available.  Cohorts with prior COVID-19 infection improve vaccine outcomes (https://www.nature.com/articles/s41541-022-00546-1; https://www.nejm.org/doi/full/10.1056/NEJMc2214916) or healthy subjects having better vaccine responses than patients (https://pubmed.ncbi.nlm.nih.gov/35693807/) or AB titers at vaccination 1 could predict the outcome of 2nd vaccination or correlation between adverse reaction and vaccine outcome (https://www.ncbi.nlm.nih.gov/pmc/articles/PMC9412348/). Hence, it becomes less interesting to the readers also given the fact that now bivalent boost has been approved lack of such analysis after the bivalent boost seems less relevant to the field.

1. It helps to provide/display overall responses and then stratify subjects with different titers for better understanding.  Also, separating infected and uninfected from the HCW cohort.

2. Cohorts to be defined well (many details are poorly described). For eg why some HCWs required re-evaluation (line 143)? what was the problem? booster - third dose? why were most of the analyses done 4 months after the second dose as the responses start declining after 90 days, in other words, peak response analysis was not done and compared.

3. why HCW with an infection history, was still included in the analysis as the higher median of Ab titers observed could be highly influenced by the infected donors within HCW

Reviewer 2 Report

The monitoring  of both the humoral and the cellular immune responses against the SARS-CoV-2 Spike protein in HCWs and different categories of frail  patients, including cancer patiens,  after  the vaccination schedule has been addressed in several papers.  . If the primary aim was to demonstrate a direct correlation between antibody level and T-cell response, this objective was quite predictable for healthy subjects and as regards cancer patients it is also predictable that the level of response B and T depend on the impact of the disease on the immune system.

The aspect that I find interesting and characterizing this research regards the deepening of the  relation between the manifestation of adverse reactions and the extent of the  immune response, in particular   the underlying association between  side effects and cellular response.

The data of this paper suggest  that the presence of vaccination side effects might reflect a stronger cell-mediated immune response, but  further investigations would be useful to confirm this preliminary evidence. Based on this, did the authors consider assessing the levels of other cytokines produced upon specific activation such as TNFa and IL2?

The polyfunctionality of the cell-mediated response is an important aspect of the efficacy of a vaccine and therefore the measurement of the single cytokine is reductive, since, for example, the levels of IL2 production are strictly correlated to the long-term maintenance of the immunological memory.

Reviewer 3 Report

A small but decisive well designed, well conducted and well discussed study. Strongly appropriate the definition of the two study groups. Particularly interesting the discussion about the different time trends of the humoral and the cellular arms of the immune responses induced by both vaccination and infection. Just a suggestion: in my opinion, useful to add a short mention of the time period during which the study was conducted, so that some consideration could be made about a possible relationship of the results with the intervening epidemiological situation of the pandemic in Italy (incidence, prevailing SARS-Cov-2 lineages, population coverage by the vaccination campaign). Clearly declared the sources of possibile conflicts of interest, certainly non afflicting the fairness and the soundness of the study.

Reviewer 4 Report

Very well written manuscript, underlining points of strength and limitations.

I suggest some minor revisions as follow:

1)     Check spelling at line 27 (influence on)

2)     The durability of vaccine immune response is also reported for B cell response, and this consideration can be included in the work. Other manuscripts reporting data on healthy people can be cited (ie. DOI: 10.3389/fimmu.2021.740708 , DOI: 10.1126/sciimmunol.abi695)

3)     The problem of therapeutic immunosuppression of frail subjects is common for different disease, also for frequent cases of co-morbidities. It should be useful to cite other published works at line 89 to give an overview of the effect of immunosuppressive treatment on humoral and cellular immune response specific for  SARS-CoV-2 (ie doi:10.3389/fimmu.2022.1017863,  doi.org/10.1182/bloodadvances.2021006599, doi:10.2169/internalmedicine.9223-21)

4)     At line 108, specify the “low level of IgG” against S-RBD?

5)     Define “cancer patients” for the fist time they are cited

6)     Tab 1, include % value for “previous SARS-Cov-2 infection”

7)     Clarify how the previous infection of subjects involved in the study was monitored (test on swab, declaration of subjects, …)

8)     Fig. 4, include the title of y axis

9)     Line 369, check spelling (maybe “booster”)

10)  In the discussion, define “the end of vaccination schedule”, since it is variable